# Macrophages and Intervertebral Disc Degeneration

**DOI:** 10.3390/ijms24021367

**Published:** 2023-01-10

**Authors:** Jinsha Koroth, Erick O. Buko, Rebecca Abbott, Casey P. Johnson, Brenda M. Ogle, Laura S. Stone, Arin M. Ellingson, Elizabeth W. Bradley

**Affiliations:** 1Department of Orthopedic Surgery, School of Medicine, University of Minnesota, Minneapolis, MN 55455, USA; 2Department of Veterinary Clinical Sciences, College of Veterinary Medicine, University of Minnesota, St. Paul, MN 55108, USA; 3Center for Magnetic Resonance Research, University of Minnesota, Minneapolis, MN 55455, USA; 4Department of Rehabilitation Medicine, School of Medicine, University of Minnesota, Minneapolis, MN 55455, USA; 5Department of Biomedical Engineering, College of Science and Engineering, University of Minnesota, Minneapolis, MN 55455, USA; 6Stem Cell Institute, University of Minnesota, Minneapolis, MN 55455, USA; 7Department of Anesthesiology, School of Medicine, University of Minnesota, Minneapolis, MN 55455, USA

**Keywords:** Tgfβ, inflammation, spine, macrophage polarization, hypoxia, low back pain

## Abstract

The intervertebral disc (IVD) aids in motion and acts to absorb energy transmitted to the spine. With little inherent regenerative capacity, degeneration of the intervertebral disc results in intervertebral disc disease, which contributes to low back pain and significant disability in many individuals. Increasing evidence suggests that IVD degeneration is a disease of the whole joint that is associated with significant inflammation. Moreover, studies show elevated macrophage accumulation within the IVD with increasing levels of disease severity; however, we still need to understand the roles, be they causative or consequential, of macrophages during the degenerative process. In this narrative review, we discuss hallmarks of IVD degeneration, showcase evidence of macrophage involvement during disc degeneration, and explore burgeoning research aimed at understanding the molecular pathways regulating macrophage functions during intervertebral disc degeneration.

## 1. Introduction

Low back pain (LBP), one of the most prevalent musculoskeletal conditions, has an estimated lifetime occurrence of 70–85% and often results in significant disability and financial burden [1,2]. Although causes of LBP are often unclear, the intervertebral disc (IVD) is a primary source of LBP, with an estimated 34–68% of patients classified with discogenic pain, or pain due to intervertebral disc degeneration [3]. Signs of degeneration are very common in the general population (upwards of 70% of adults) [4], and do not always relate directly to symptomology [5]. However, IVD degeneration is more prevalent in those with LBP [6].

The IVD enables the spine to undergo extensive multidirectional motion as well as absorb, dissipate, and transmit energy along the spinal column. These capabilities are due to the complex interaction between two morphologically, biomechanically, and biochemically distinct tissues: the annulus fibrosus (AF) and the nucleus pulposus (NP). Two cell types of different developmental origins primarily compose the IVD, chondrocyte-like cells within the nucleus pulposus and the inner annulus fibrosis, and fibroblast-like cells within the outer AF; however, single-cell RNA sequencing studies suggest additional cellular heterogeneity and the potential for IVD progenitor cells within model organisms [7,8,9]. The AF consists of highly organized concentric rings (lamellae) of fibrocartilaginous material that surround the NP. Type I collagen fibers with altered directions primarily comprise lamellae within the AF [10]. In contrast, a hydrated, disorganized matrix rich in proteoglycans and type II collagen constitutes the NP [10,11]. Maintenance of the NP and AF relies on nutrition exchange through the cartilaginous and vertebral endplates, since healthy adult IVDs are only superficially innervated and are predominantly avascular [12,13,14].

During IVD degeneration, the disc becomes more disorganized and the distinction between the AF and NP becomes less apparent [15]. Degradation and loss of proteoglycans directly reduce the hydration capacity of the IVD and osmotic pressure [16,17]. With advancing degeneration, alterations in collagen structure, type, and distribution also typify changes to the IVD, albeit not as significantly as the changes in proteoglycans [18,19]. These changes in the macromolecular structure impact IVD function, resulting in diminished mechanical integrity, competency, and disc height [16].

IVD degeneration exhibits a complex etiology. Once thought to result from simple “wear and tear”, we now appreciate the multiple underlying causes of IVD degeneration. Mechanical factors, including compressive loading, shear stress, and vibration, can all contribute to disc degeneration [20]. In addition, structural failure (e.g., annular tear, bulging disc) also associates with IVD degeneration [20,21]. Inadequate metabolite transport across the end plate, aging, genetics, and metabolic disorders (e.g., obesity, diabetes) may likewise contribute to disc degeneration [20,22,23].

Several potential mechanisms of IVD degeneration may contribute to chronic LBP (Figure 1). First, reduced disc height, disc bulging, and/or disc herniation secondary to matrix changes can result in mechanical compression of sensory neurons in the dorsal root ganglia or spinal nerve roots as they exit the spinal column; this is a driver of neuropathic radiating leg pain. Specifically, pressure against the sciatic nerve leads to substantial radiating leg pain [24]. Furthermore, increased nerve growth factor (NGF) expression in IVD degeneration links to increased pain-generating innervation [25,26,27]. Second, as discs degenerate and lose structural integrity, the inflammatory environment is no longer contained within the disc and can have a collateral effect on nearby nerve fibers, potentially increasing pain [25,28]. Third, spinal instability and local inflammation secondary to IVD degeneration may result in inflammation and mechanical strain in adjacent tissues such as facet joints, ligaments, and spinal muscles [29,30,31]. Finally, increased vascularization and neural ingrowth accompany IVD degeneration; some of these nerves are nociceptive [32,33]. As a result, nociceptors can respond to mechanical forces and biochemical agents (e.g., Tumor necrosis factor (TNF)-α, low pH) within the IVD that are usually not detected [29]. These changes are exacerbated by increased expression of pro-inflammatory (e.g., Interleukin (IL)-1β, TNF-α) and pro-nociceptive (e.g., NGF) mediators by cells within the IVD as they degenerate, including factors produced and released by macrophages [27].

Macrophages regulate many of the biological processes that occur during disc degeneration. Below, we discuss how macrophages are derived and maintained, shedding light on the potentially different functions of macrophages of alternate developmental origins. We also summarize the literature showing that levels of macrophages within the disc increase along with IVD degeneration and touch on how macrophages may interface with other tissue types (e.g., nerves, bone) within the IVD. We also summarize what is known about regulatory mechanisms governing macrophage chemotaxis, production of inflammatory cytokines and chemokines, as well as secretion of matrix degrading enzymes. Although we are beginning to understand the molecular regulation of macrophages during IVD degeneration and back pain, as summarized here, there are clear gaps in our knowledge that we address herein.

## 2. Myeloid Lineage Cells and Their Derivatives

Production of macrophages within the body occurs via two distinct developmental origins. The first mechanism occurs through hematopoietic stem cell (HSC)-mediated generation. HSCs give rise to common myeloid progenitors (CMPs) and common lymphoid progenitors (CLPs) during hematopoiesis. While natural killer cells as well as B and T lymphocytes differentiate from CLPs, myeloid lineage cells (e.g., granulocytes, megakaryocytes, dendritic cells, and monocytes) derive from CMPs. Monocytes serve as the direct progenitors of macrophages generated from HSCs. Monocytes egress daily from the bone marrow in a circadian fashion, a process enhanced by inflammation [34]. This produces two types of monocytes within the circulatory system, classified by the relative levels of the surface marker Lymphocyte antigen 6 complex (Ly6c) in mice, with cluster of differentiation (Cd) 14 serving as the human equivalent. Circulating monocytes patrol the circulation, and low Ly6c surface expression (e.g., Ly6c^Low^ cells) characterizes these cells. In contrast, migratory monocytes (e.g., Ly6c^High^ cells) cross the vascular endothelium into tissues [35]. Migratory monocytes have a half-life of approximately one day once within the circulation; these cells can either convert into circulating monocytes or exit the vascular system to function as antigen-presenting cells such as macrophages or tissue monocytes (Figure 2) [35].

The second mechanism of macrophage generation occurs during embryonic development. These macrophages are not HSC-derived but trace their origins to the fetal yolk sac and liver (Figure 2) [36]. Known as tissue-resident macrophages (TRMs), these cells are the only immune cells present prior to bone marrow cavity formation and HSC-mediated hematopoiesis [36]. These embryonic-derived macrophages are long-lived and have a high capacity for self-renewal [37]. TRMs, ranging from Kupffer cells within the liver to microglia within the central nervous system to Langerhans cells within the skin, and osteoclasts within bone, populate virtually every tissue within the body [38]. Despite this, we know little about the presence and potential functions of TRMs within the intervertebral disc.

## 3. Macrophage Function and Polarization

Metchnikoff first described macrophages during the late 19th century; the term macrophage derives from the Greek words makros for “large” and phagein meaning “to eat” [39]. Macrophages function broadly within the innate immune system to protect organisms against antigens [39]. As part of the innate immune system, macrophages clear foreign objects from tissues, including pathogens and implant wear particles released from joint implants, through an engulfment process called phagocytosis [40]. Although activity can change, all macrophage subsets exhibit phagocytic activity. During this process, macrophages surround a large particle or microbe with their plasma membrane [39]. Once engulfed, this lipid bilayer-surrounded particle is known as a phagosome [39]. Fusion of the phagosome with lysosomes within the macrophage allows for digestion of the phagocytosed antigen [39]. In addition to inert particles and microbes, macrophages also phagocytose and remove dysfunctional cells and cellular debris in a process known as efferocytosis [39]. This includes apoptotic, cancerous, and senescent cells within the body [41,42]. Macrophages also clear byproducts of matrix degradation (e.g., collagen fragments) during the tissue regeneration process [43]. Partially due to this later function, macrophages are essential for tissue regeneration in model organisms [44,45,46].

Macrophage-colony stimulating factor (M-CSF) and IL-34 induce the differentiation and survival of macrophages [47,48], a process transcriptionally controlled by factors such as runt-related transcription factor (Runx) 1, PU.1, and CCAAT-enhancer binding protein (CEBP)α [49,50]. Macrophages express various pattern-sensing molecules, including scavenger receptors, pattern recognition receptors, and nuclear hormone receptors [51]. In addition, cytokine and chemokine receptors facilitate the adaptation of macrophages to the local milieu and changing conditions [51].

Although highly heterogeneous and plastic by nature, the M0-M1-M2 classification system generally helps delineate different functions of macrophages (Figure 2). M0 represents unpolarized macrophages [52]. M1 act as inflammatory mediators, whereas M2 promote tissue healing and homeostasis [52,53]. Macrophages polarize to the M1 state in response to inflammatory stimuli, including bacterial-derived lipopolysaccharides (LPS), interferon (IFN)-γ, and TNF-α [52]. Transcriptional activity of activator protein 1 (Ap-1), hypoxia-inducible factor (HIF)-1-α, nuclear Factor (NF)-κB, stress associated endoplasmic reticulum protein 1 (SERP-1), and signal transducer and activator of transcription (STAT)1/3 facilitate the attainment of M1 polarization [54]. Once polarized, M1 produces cytokines including IL-1β, IL-6, IL-12, IL-23, IFN-γ, TNF-α, and C-X-C chemokines to promote inflammation [55,56,57,58,59,60,61,62]. Furthermore, iNOS induction promotes reactive oxygen species generation to enhance the M1 inflammatory response further. The generation of reactive oxygen species (ROS) by macrophages is also accomplished through alterations to cellular metabolism to facilitate the inflammatory response. M1 increase their rate of cellular glycolysis and utilize the pentose phosphate pathway for ROS generation [54]. Additionally, breaks in the citric acid cycle generate additional ROS [54].

While M1 polarize in response to and promotes inflammation, M2 supports tissue repair and inflammatory resolution. IL-4 and IL-13 induce M2 [63], with transcriptional activity of STAT6 and interferon regulator factor (Irf) 4 facilitating this change in phenotype [64,65]. M2 polarized macrophages produce several cytokines that promote tissue anabolism including IL-10, transforming growth factor (Tgf)β1, bone morphogenetic protein (Bmp) 2 and osteopontin/SPP1 and promote collagen deposition [66,67,68]. Arginase-1 (Arg1), C-type mannose receptor 1 (Mrc1, Cd206), macrophage galactose-type C-type lectin (Mgl), resistin-like alpha (Retnla), chitinase-3-like protein 3 (Chi3l3), glutamine synthase (Glu1), and Cd163 are amongst some of the genes and surface markers that phenotypically characterize M2 [69]. Increased Arg1 activity and Glu1 expression both facilitate increased citric acid cycle activity by providing glutamine as an additional substrate [54].

## 4. Roles of Macrophages in Disc Degeneration

In addition to their phagocytic activity and antigen-presenting functions, macrophages are critical regulators of the tissue healing process; thus, altered function is associated with many degenerative diseases, including IVD degeneration. Below, we discuss the association of macrophages with disc degeneration as well as studies aimed at understanding how macrophages regulate pain and extracellular matrix (ECM) degradation accompanying IVD degeneration.

### 4.1. Increased Vascularization May Facilitate Immune Cell Invasion during Disc Degeneration

Healthy adult IVDs were historically viewed as immune-privileged, avascular joints, but studies demonstrate the presence of blood vessels within the outer annulus fibrosis of normal adults [13]. This raises the potential for immune cell invasion into the healthy adult IVD [70]. There is also some disagreement about the extent of vascular invasion associated with disc degeneration. A prior study showed limited ingrowth of vasculature correlating with disc degeneration, but this was limited to immunohistochemical staining (e.g., PECAM) of tissues [14]. In a recent study, ingrowth of vasculature to the endplate and inner layers of the annulus fibrosus of cadaveric human spines was associated with degenerative discs in humans [12,13]. This increase in vascularization may facilitate the ingress of immune cells into degenerating IVDs. Regardless of the extent of vascular invasion, a chronic inflammatory state and elevated levels of pro-inflammatory cytokines (e.g., IFN-γ, IL-1α, IL-1β, IL-6, IL-17, and TNF-α) and chemokines (e.g., CC motif chemokine ligand (CCL) 2, CCL3, and CXCL10) produced by IVD cells characterize the degenerative process in multiple species (Figure 3) [71,72]. These inflammatory cytokines promote a catabolic response, resulting in extracellular matrix loss, cellular apoptosis, production of neurotrophins, and infiltration of immune cells, including macrophages, into the disc [71]. Recruitment of immune cells to the disc, including macrophages, amplifies the inflammatory response [71].

### 4.2. Macrophage Levels Associate with Degenerative Disc Disease Severity

Macrophages infiltrate into degenerated intervertebral discs in humans and model organisms [73,74,75]. In humans, levels of macrophage markers positively associate with disc degeneration within the nucleus pulposus and endplate, with stronger associations within unhealthy regions that exhibited structural compromise of the disc in cadaveric specimens [73]. Moreover, the prevalence of CC motif chemokine receptor (CCR) 7^+^ macrophages (i.e., M1) as well as Cd163^+^ (e.g., M2) macrophages increases with age and degeneration [73]. In addition, macrophages within the disc exhibited co-staining for M1 and M2 markers (e.g., one cell expressed both M1 and M2 surface markers), suggesting this binary view of macrophage phenotypes may be too simplistic [73]. In contrast, cells expressing the M2 surface marker Mrc1/Cd206 did not associate with disc degeneration in humans [73].

Murine injury models also support the association between macrophages and disc degeneration. In a murine puncture injury-induced model of lumbar IVD degeneration, observed infiltration of F4/80^+^ macrophages within the IVD occurred immediately after injury (day 4), and persisted within the disc for up to 12 months following injury [76]. Subsets of macrophages within the disc were detected following puncture injury in mice, with M1 markers (e.g., TNF-α, IL-1β, Nos2 mRNA) prevalent early in response to injury. In contrast, M2 markers (e.g., Chi3l3, Tgfβ, Cd206 mRNA) increased over the course of 28 days; however, confirmation of disc degeneration and the extent of induced damage were not evaluated in this study [77]. Co-culture models of degenerated IVDs and macrophages also suggest that IVD-produced IL-1β polarizes macrophages to a pro-inflammatory phenotype [78]. Recent studies also suggest that exercise limits pain and macrophage-mediated inflammation within the disc in mouse models [79]. Broad clearance of all macrophage subsets in vivo via clodronate liposomes limits production of inflammatory cytokines when Cd11b^+^ cells from the disc are cultured in vitro [80], but an examination of the effects of macrophage clearance on disc degeneration was not described.

Single-cell RNA sequencing studies also demonstrate the presence of macrophages within degenerating IVDs. A single cell-sequencing analysis of the nucleus pulposus from human IVD degeneration specimens demonstrated a significant interaction between subsets of nucleus pulposus cells and macrophages not observed in normal controls, demonstrating that degenerated discs recruit macrophages to the nucleus pulposus [81]. Moreover, a subsequent study found that nucleus pulposus progenitor cells showed an interaction with macrophages and that macrophage polarization impacts IVD cellular metabolism [82]; however, functional studies confirming the authors’ observations are needed. Experiments in which inducible clearance of macrophages and/or macrophage subsets (e.g., MAFIA: LysMCre^ERT^ or Cd169-Cre) follows disc injury would substantially increase our understanding of the function of macrophages during the course of disc degeneration. Likewise, the functions of macrophage subsets during disc degeneration are also unclear.

### 4.3. Intervertebral Disc-Bone Crosstalk

Intervertebral disc degeneration is also associated with calcification if the cartilaginous endplates and subchondral bone (i.e., calcified endplate) change within the vertebrae. In human cadaveric specimens, higher grades of disc degeneration were accompanied by increased subchondral bone volume fraction (e.g., bone volume/total volume) and enhanced trabecular thickness [83]. Modic changes to vertebral subchondral bone observed with magnetic resonance imaging may also reflect inflammation leading to disc degeneration [84]. Disc degeneration is also closely related to osteophyte formation within the spine [85]. These factors also promote disease progression and contribute to pain severity [85]. Increased numbers of osteal tissue macrophages (i.e., osteomacs) are also observed with subchondral bone sclerosis, which is mediated by increased oncostain M-mediated osteoblast differentiation [86]. Macrophages within the IVD may also influence bone resorption through the production of pro- and anti-inflammatory cytokines [87,88].

### 4.4. Regulation of Pain Mediation by Macrophages within Degenerated Discs

Nerve fibers are present within healthy adult discs [29], but significant nerve ingrowth is associated with disc degeneration [28,89,90]. Nerve infiltration into the IVD is associated with increased levels of NGF, brain-derived neurotrophic factor (BDNF), and substance P [25,26,28,91]. While pain is often associated with disc degeneration [92,93,94], signs of IVD degeneration on imaging are also present in pain-free individuals [95,96,97,98]. Thus, a critical need to identify mechanisms promoting pain during disc degeneration exists.

Neuroimmune crosstalk, including from inflammatory factors produced by macrophages, promotes pathological pain and may be a mechanism of pain mediation during IVD degeneration [99]. Macrophages facilitate pain by producing a battery of gene products, including prostaglandin 2, NGF, and nitric oxide, as well as the inflammatory cytokines and chemokines noted above [100,101,102]. In an organ culture model, macrophages within the IVD produced TNF-α, cyclooxygenase (Cox) 2, and IL-8 [103]. Moreover, the authors demonstrate that neutralization of TNF-α or IL-8 limited mechanical allodynia (e.g., pain) in a rat model of disc degeneration [103], and inhibiting IL-8 signaling in a mouse model reduced both behavioral signs of low back pain and IVD inflammation [104]. In addition, when Cd14^+^ cells were isolated from degenerated human IVDs and stimulated with TNF-α and IL-1β, production of the pain-related molecules Calcitonin gene related peptide CGRP and NGF increased [105]. Likewise, infiltration of macrophages into the sciatic nerve and dorsal root ganglia near the degenerated IVD has been observed [106], suggesting that direct neuroimmune interactions may contribute to the pain associated with disc degeneration. To date, the majority of studies associate macrophage pain-mediator production with pain during disc degeneration; thus, there is a critical need to address the role of various macrophage-produced pain mediators in discogenic pain.

### 4.5. Extracellular Matrix Degradation

Macrophages participate in the degradation and remodeling of the extracellular matrix (ECM) through their production of metalloproteinase (MMP)s and A disintegrin and metalloproteinase with thrombospondin motifs (ADAMTSs); thus, macrophages may participate in critical pathological mechanisms during disc degeneration because of the importance of the ECM integrity to disc health/homeostasis. Human genetic evidence supports the importance of ECM integrity, as several genetic variants of ECM genes are associated with IVD degeneration (Table 1). Genes encoding the collagen isoforms collagen (Col1) 1a1 and Col9a3, produced by cells within the NP and AF, are associated with increased susceptibility to intervertebral disc disease [107,108,109]. Coding variants for extracellular matrix proteins, including CAP-Gly domain-containing linker protein 1 (CILP), asporin, and aggrecan have also been identified in individuals with lumbar disc disease [110,111,112,113]. Likewise, mutations in genes encoding matrix degrading enzymes, including ADAMTS-4/5 and MMPs, as well as alterations in the gene encoding thrombospondin 2 [114,115,116].

Studies aimed at understanding the role of macrophages in controlling extracellular matrix degradation during intervertebral disc degeneration have yielded mixed results. Early studies showed that macrophages induced MMP-3 production by cells isolated from the IVD [117]. TNF-α induced ECM degrading enzymes such as ADAMTS-4/5, and MMP-13 limited synthesis of aggrecan and Col2a1 [118]. This effect was mitigated by soluble factors produced by M2 polarized cells in vitro [118]. The authors also demonstrated that M2 limited the effect of TNF-α in an IVD organ culture model [118]. Although not direct evidence supporting the role of macrophages in ECM degradation, transgenic expression of hTNF-α resulted in spontaneous annular tears associated with immune cell invasion, but without changes to expression of aggrecan and collagen [119]. In contrast, soluble factors derived from IL4-stimulated THP-1 cells (i.e., an immortalized macrophage cell line) promoted expression of MMP-3 and MMP-9 concomitant with reduced Aggrecan and Collagen II production by nucleus pulposus cells [120].

## 5. Molecular Control of Macrophage Function during IVD Degeneration

Although a number of studies show a positive correlation between (1) macrophages within the IVD, (2) disc degeneration, and (3) associated pain, we still have much to learn about the control of macrophage function within the disc environment. Below, we summarize current knowledge of how chemokine signaling affects macrophage migration into degenerated discs, as well as how inflammation, Tgfβ signaling, and hypoxia control macrophage function during IVD degeneration.

### 5.1. Chemokine Signaling and Macrophage Migration

Chemokines are a large family of small, secreted proteins that act as chemoattractants to increase cellular migration. Nucleus pulposus cells produce an array of chemokines, including Ccl2 (Mcp-1), Ccl7, and Ccl8, and their expression levels connect with the severity of disc degeneration [121] (Figure 3); chemokine serum levels likewise associate with disease severity [122]. Induced by the production of IL-1β and TNF-α within the degenerated disc, chemokine signaling increases macrophage tissue infiltration [123]. Specifically, inflammatory cytokine production by nucleus pulposus cells leads to enhanced expression of Ccl3 by nucleus pulposus cells, which promotes macrophage migration via Ccr1 [124]. In support of the role of Ccr1 in promoting macrophage infiltration during IVD degeneration, blocking Ccr1/2 in a rabbit annular puncture model decreased disc degeneration, limited inflammation, and blocked macrophage migration in vitro [125]. Ccl2 induction following injury in a mouse model of IVD degeneration also enhanced macrophage localization [126]. Studies also suggest that resistin acts via Toll-like receptor (TLR) 4 to activate p38 mitogen-activated protein kinase (MAPK) and NF-κB-dependent induction of Ccl4, leading to increased macrophage infiltration [127] (Figure 3). Moreover, specific p38 MAPK isoforms may be critical to macrophage functions within the disc [128,129]. Despite these studies, thorough analyses each identified chemokine’s requirement would vastly improve our knowledge of macrophage migration into the disc during degeneration, as well as the functional consequences on disease progression.

### 5.2. Macrophage Inflammasome Activation during IVD Degeneration

Inflammasomes are multiprotein complexes of innate immune receptors and sensors that facilitate caspase-dependent inflammation and cell death. Traditionally, inflammasomes are activated within innate immune cells, including macrophages, but other cell types are also reported to induce inflammasome activation [130]. Mounting evidence supports a functional role for the inflammasome during intervertebral disc degeneration (Figure 3) [131,132]. Inflammation-inducing stimuli, including endogenous danger/damage-related molecular patterns (DAMPs) and exogenous pathogen-related molecular patterns (PAMPs), induce inflammasome assembly, leading to the activation of alternative caspases (e.g., caspase-1) and downstream cellular responses including increased inflammatory cytokine expression, pyroptosis, and apoptosis [133,134]. Increased inflammation, pyroptosis, ECM loss, and death of IVD cells all occur in response to activation of the NLR family pyrin domain containing (NLRP) 3 inflammasome in IVD degeneration [135]. Many inflammasome-related genes and proteins have been identified, but the functions of NLRP3 are best characterized in IVD degeneration [136]. In humans, levels of IL-1β, caspase-1, and NLRP3 positively associate with IVD degeneration severity [132,137,138], as well as in a rat model of IVD degeneration [137]. Bai et al. also show that NLRP3 facilitates pyroptosis of human nucleus pulposus cells [131]. Genetic evidence also suggests that pyroptosis of nucleus pulposus cells is associated with low back pain in humans [139,140]. Moreover, inflammasome activation links to endoplasmic reticulum stress, mitochondrial dysfunction, and reactive oxygen species-mediated damage seen in IVD degeneration. Degradation of the IVD extracellular matrix is also thought to promote inflammation and activation of the inflammasome [141], but the requirement of specific cell types, be they invading immune cells or nucleus pulposus cells, to alter disc cell biology and disc degeneration is poorly defined.

The inflammasome may also be activated by advanced glycation end-products (AGEs) present within degenerative discs [142]. In vitro studies on human myeloid cells treated with AGEs revealed a substantial increase of NLRP3, cleaved caspase-1, pro-IL-1β, and IL-1β [143]. Additionally, mechanistic studies show that NLRP3 inflammasome activation occurs via mitochondrial dysfunction and ROS generation, both of which are induced by the accumulation of AGEs in nucleus pulposus cells [144]. Studies demonstrating that exogenous TNF-α promotes NF-κB-dependent mitochondrial dysfunction and ROS generation, leading to NLRP-containing inflammasome activation by nucleus pulposus cells further support this notion [145]. Bai et al. show that delivering a ROS-scavenging hydrogel loaded with rapamycin can limit inflammation and enhance M2 polarization within the disc [146].

### 5.3. Tgfβ Signaling

The pleiotropic effects of Tgfβ exert both harmful and beneficial effects within numerous tissues and disease states. Mutations to Tgfβ signaling components Small mothers against decapentaplegic (SMAD) 2/3 are associated with osteoarthritis, intervertebral disc abnormalities, and degeneration [147,148] (Table 1). These mutations not only support the importance of Tgfβ signaling in human disease but also may affect the functions of macrophages during IVDD and/or how other cell types within the disc respond to macrophage-produced signals. Levels of Tgfβ within the disc positively align with disc degeneration severity in humans [149,150,151,152], but it is unknown if this is a cause or consequence of disc degeneration, and the effects on specific cells within the disc are likewise unclear. Recent studies suggest that Tgfβ polarizes macrophages towards an M2 phenotype, but the change in macrophage activation in response to Tgfβ may be distinct from that elicited by IL-4/IL-13 [153,154,155]. To evaluate levels of M2 within murine intervertebral discs, Kawakubo et al. utilized Cd206 as a marker for tissue resident macrophages. They demonstrated that Cd206^+^ cells reside within the disc, but administration of Tgfβ to the mice (e.g., intraperitoneal injection) enhanced levels of Cd206^+^ cells in both young and aged mice [156]. In contrast, systemic delivery of a Tgfβ inhibitor had the opposite effect; however, it is unknown if this effect was localized to the disc or due to systemic effects of Tgfβ. In addition, several studies demonstrate that levels of Tgfβ and TgfβRII within the IVD decline with age in rodents; thus, Tgfβ may play a role in the maintenance of the IVD [156,157]. In addition, studies on human specimens showed that exogenous Tgfβ treatment reduces IVD inflammation both in vitro and *in vivo*. Given the contrasting roles of Tgfβ during disc degeneration, additional work is needed to clarify the cell-specific effects of Tgfβ during IVD degeneration to better understand the mechanisms of this pathway during disease progression.

### 5.4. Hypoxia and Oxidative Stress

Macrophages exhibit large changes in gene expression to facilitate adaptation to hypoxic conditions; much of this adaptation occurs through HIF-1-α-dependent mechanisms [158,159]. In addition, HIF-1-α also coordinates polarization of macrophages within normoxic conditions [160,161]. The intervertebral disc lacks significant vascular supply; thus, the nucleus pulposus is hypoxic. Despite the known correlation between disease severity and macrophage accumulation within degenerated discs, we know little about the link between changing oxygen tensions and the potential effects of macrophages on disc degeneration. Observational studies show that deletion of HIF-1-α using sonic hedgehog (Shh)-cre to target the IVD results in accelerated disc degeneration [162], suggesting that nucleus pulposus intrinsic responses to hypoxia critically maintain the disc. Moreover, exogenous expression of HIF-1-α limits disc degeneration and reduces apoptosis of nucleus pulposus cells in a murine disc injury model [163]. Likewise, conditional deletion of HIF-2-α limits age-dependent changes within the IVD [164]. There are a few reports showing that hypoxia impacts the response of nucleus pulposus cells to macrophage-produced inflammatory cytokines. For instance, when cultured in hypoxic conditions, inflammation, chemotaxis, matrix degradation, and angiogenesis of nucleus pulposus cells are associated with exposure to IL-1β, IL-20, and Bmp-2 [165], but the specific role of these cytokines to the effects of changing oxygen tension within the disc, as well as the cellular mediators, requires further study. IL-1β increases expression of HIF-1-α by nucleus pulposus cells cultured in hypoxia, concomitant with increased production of MMP-1 and diminished TIMP-1/2 levels; however, factors present within macrophage-conditioned medium inhibited the pro-inflammatory response of hypoxia-cultured nucleus pulposus cells [166].

## 6. Conclusions and Future Directions

Intervertebral disc degeneration leads to chronic back pain, for which there are few treatment options; thus, understanding the degenerative process will aid the intelligent design of therapeutic strategies to promote disc regeneration and mitigate pain. In its end stage, IVD degeneration results in joint space narrowing (e.g., reduced disc height), adjacent vertebral bone sclerosis, and osteophyte formation, resulting in spinal OA. Associations between macrophage numbers within the disc and inflammatory phenotypes with disease severity have been established, but we lack definitive knowledge demonstrating either a causative or a consequential role for macrophages during IVD degeneration. Understanding how influences such as age, BMI, and other drivers of degenerative processes affect macrophage subsets and further defining potential functional roles for macrophages in IVD degeneration is paramount to our ability to limit disease progression.

## Figures and Tables

**Figure 1 ijms-24-01367-f001:**
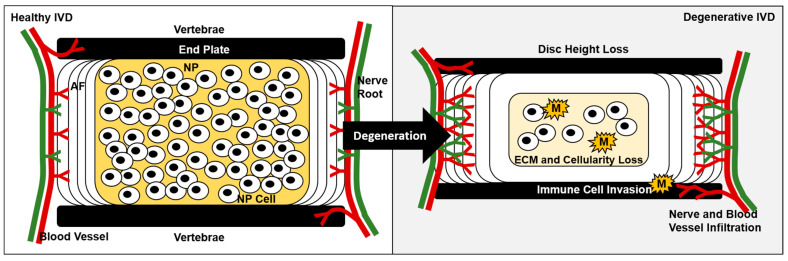
Changes associated with intervertebral disc degeneration. Disc height loss, ECM degradation, nerve and blood vessel infiltration, inflammatory cell invasion, and loss of cellularity accompany degenerative disc disease.

**Figure 2 ijms-24-01367-f002:**
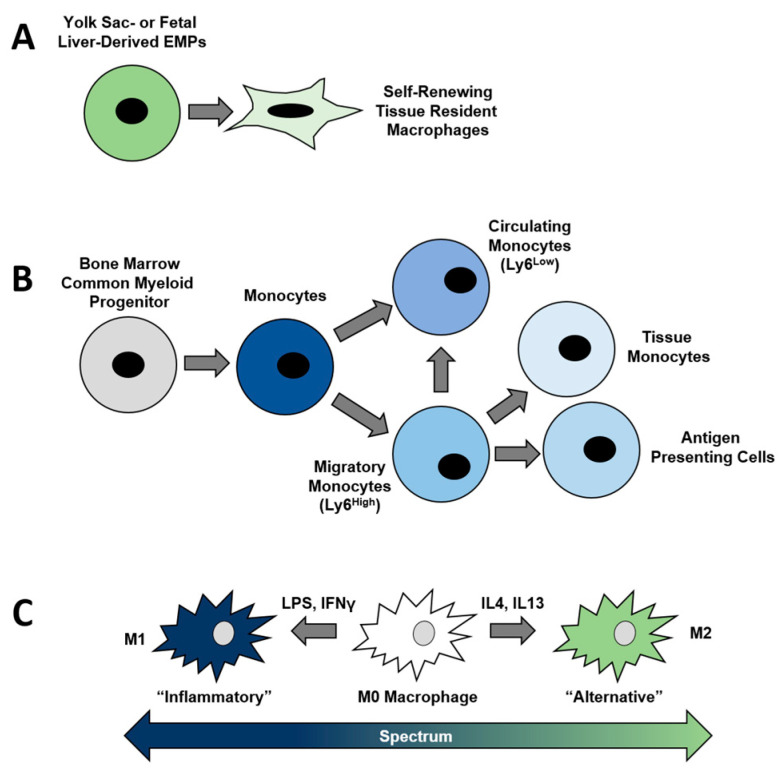
Origins and functional polarization of macrophages. (**A**) Tissue resident macrophages derive from the fetal yolk sac and liver. (**B**) Hematopoietic stem cell (HSC)-derived common myeloid progenitors give rise to myocytes and macrophages. (**C**) Inflammatory stimuli such as LPS and IFN-γ polarize macrophages towards and M1 phenotype, whereas IL-4 and IL1-3 promote M2 polarization.

**Figure 3 ijms-24-01367-f003:**
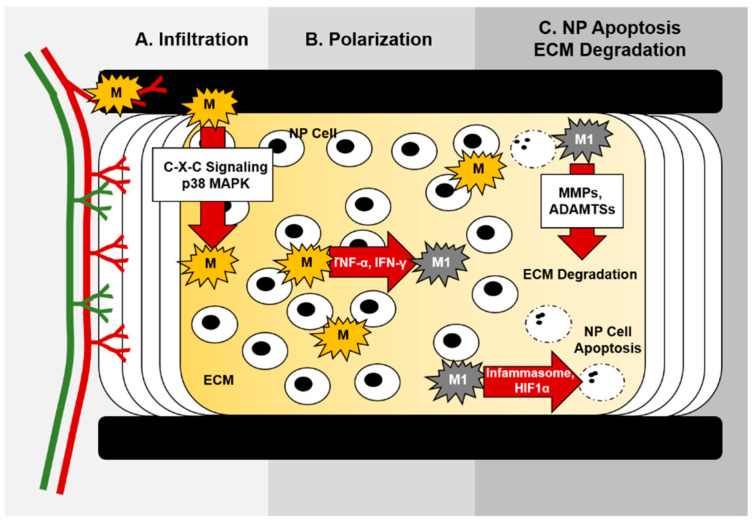
Molecular control of macrophages during intervertebral disc degeneration. (**A**). C-X-C and p38 MAPK signaling promote macrophage infiltration into the IVD during degeneration. (**B**). TNF-α and IFN-γ within the local microenvironment promote M1 polarization during IVD degeneration. (**C**). Inflammasome activation and HIF-1-α signaling influence nucleus pulposus cell apoptosis, and macrophage-produced MMPs and ADAMTSs promote ECM degradation within degenerated discs.

**Table 1 ijms-24-01367-t001:** Known genetic mutations that are associated with lumbar disc disease. We searched the OMIM Database for genetic variants that are associated with intervertebral disc degeneration.

Gene	OMIM	Clinical Disease
*ACAN*	155760	IVD Degeneration
*ASPN*	608135	IVD Degeneration
*CHAD*	602178	IVD Degeneration
*CHST3*	603799	SEDCJD
*CILP*	603489	IVD Degeneration
*COL9A3*	120270	IVD Degeneration
*COL11A1*	120280	IVD Degeneration
*SMAD2*	601366	LDS6
*SMAD3*	603109	LDS3
*THBS2*	188061	IVD Degeneration

## Data Availability

The authors utilized the OMIM database during the construction of this manuscript.

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
