# Peer review of "Macrophages and Intervertebral Disc Degeneration"

_ijms, 2023, doi:10.3390/ijms24021367_

Round 1

Reviewer 1 Report

This manuscript describes the roles of macrophages in intervertebral disc degeneration, The authors provide a comprehensive review of the mechanistic pathways that contribute to the development of the disease, and associated hallmarks e.g. pain, as well as how macrophages regulate essential processes in IVD. The review is very well written and includes the essential information in this topic plus some additional evidence from human and animal studies that uncover the molecular mechanisms that control IVD homeostasis.

A few points need to be addressed:

·       Introduction, lines 67-84. Although the information is clear, the effect on the sciatic nerves which causes severe and chronic pain is not included. It is strongly suggested to refer to this feature as well along with the relative citations.

·       Section 3, elimination of cell debris by macrophages should be added.

·       Line 252, neuroimmune.

·       Section 4. In this part, a subsection describing the role of skeletal tissue in IVD degeneration would add to the review. It is known that IVD disease is closely related to the formation of osteophytes and vertebral endplate subchondral bone sclerosis which contribute to the progression of the disease as well as pain severity. In addition, macrophages secrete pro- and inflammatory cytokines that alter the balance of bone resorption and formation, locally in the IVD.

·       It is unclear how and why the authors included Table 2. A separate section/paragraph should be added for the description of the table as well as the rationale since it is referred to OA.

Author Response

We thank the Reviewers for their critical and helpful review of our narrative review on Macrophages and Intervertebral Disc Degeneration.  Below we address each comment in a point-by-point fashion.

Reviewer 1:

This manuscript describes the roles of macrophages in intervertebral disc degeneration, the authors provide a comprehensive review of the mechanistic pathways that contribute to the development of the disease, and associated hallmarks e.g. pain, as well as how macrophages regulate essential processes in IVD. The review is very well written and includes the essential information in this topic plus some additional evidence from human and animal studies that uncover the molecular mechanisms that control IVD homeostasis.

A few points need to be addressed:

  • Introduction, lines 67-84. Although the information is clear, the effect on the sciatic nerves which causes severe and chronic pain is not included. It is strongly suggested to refer to this feature as well along with the relative citations.

We added this feature of disc degeneration to our introduction, including associated citations.

  • Section 3, elimination of cell debris by macrophages should be added.

We added clearance of cellular debris and matrix degradation products along with our discussion of efferocytosis.

  • Line 252, neuroimmune.

We made this correction

  • Section 4. In this part, a subsection describing the role of skeletal tissue in IVD degeneration would add to the review. It is known that IVD disease is closely related to the formation of osteophytes and vertebral endplate subchondral bone sclerosis which contribute to the progression of the disease as well as pain severity. In addition, macrophages secrete pro- and inflammatory cytokines that alter the balance of bone resorption and formation, locally in the IVD.

We added a subsection of bone-IVD crosstalk as suggested.

  • It is unclear how and why the authors included Table 2. A separate section/paragraph should be added for the description of the table as well as the rationale since it is referred to OA.

We removed table 2.

Reviewer 2 Report

The review by Koroth et al. summarizes our understanding about the role of macrophages in contributing to disc degeneration. The manuscript is well written and requires only minor changes. While it is important to know the genetic mutations that associate with lumbar disc disease, and that identifying GO terms that are associated with spine osteoarthritis is important for IVD research, these aspects of disc degenerations are out of context in a review that focuses on the role of macrophages in disc degeneration. The manuscript would benefit from removing these aspects. The authors should expand figures 1 and 3 and provide more details. The current figures do not convey the message that is given in the figure legend and the manuscript. The conclusion should be focused more on macrophages than on genetic variances  that are associated with disc degeneration

Please find my minor comments below:

·       Page 2 Line 71: Please correct this sentence: “Furthermore, the increased expression of Nerve growth factor (NGF) in IVD degeneration links to increased innervation of pain-generating IVDs”.

·       Page 3 Lines 89-103: Please include citations to support your statements.

·       Page 4 Line 121: The statement that “These macrophages are not HSC-derived, but trace their origins to the fetal yolk sac and liver” is not shown in Figure 2. Please add/expand the figure.

·       Page 5 Line 188: The citation for “but more recent studies demonstrate the presence of blood vessels”… is from 1995. It is not new that blood vessels grow into the degenerating AF.

·       Page 7 chapter 4.4: The information about genetic variances is out of context and should be removed.

·       Several locations in the manuscript: “degenerate IVD” should be “degenerated IVD”.

·       Figures 1 and 3 don’t not show what the figure legend describe. Please add additional detail to the figures. For Figure 3 it looks as if proteins/cytokines would be outside the IVD.

·       Table 1&2 should be removed. A table summarizing the literature about macrophages in disc degeneration would be more helpful.

Author Response

We thank the Reviewers for their critical and helpful review of our narrative review on Macrophages and Intervertebral Disc Degeneration.  Below we address each comment in a point-by-point fashion.

Reviewer 2

The review by Koroth et al. summarizes our understanding about the role of macrophages in contributing to disc degeneration. The manuscript is well written and requires only minor changes. While it is important to know the genetic mutations that associate with lumbar disc disease, and that identifying GO terms that are associated with spine osteoarthritis is important for IVD research, these aspects of disc degenerations are out of context in a review that focuses on the role of macrophages in disc degeneration. The manuscript would benefit from removing these aspects.

We removed Table 2 as suggested.

The authors should expand figures 1 and 3 and provide more details. The current figures do not convey the message that is given in the figure legend and the manuscript.

We expanded Figures 1 and 3 as suggested.

The conclusion should be focused more on macrophages than on genetic variances that are associated with disc degeneration.

We revised our conclusion as suggested.

Please find my minor comments below:

  • Page 2 Line 71: Please correct this sentence: “Furthermore, the increased expression of Nerve growth factor (NGF) in IVD degeneration links to increased innervation of pain-generating IVDs”.

We corrected this sentence.

  • Page 3 Lines 89-103: Please include citations to support your statements.

We added citations accordingly.

  • Page 4 Line 121: The statement that “These macrophages are not HSC-derived, but trace their origins to the fetal yolk sac and liver” is not shown in Figure 2. Please add/expand the figure.

We expanded this figure to include embryonic-derived EMPs.

  • Page 5 Line 188: The citation for “but more recent studies demonstrate the presence of blood vessels”… is from 1995. It is not new that blood vessels grow into the degenerating AF.

We corrected this sentence.

  • Page 7 chapter 4.4: The information about genetic variances is out of context and should be removed.

Human genetic variants would affect all cell types involved with IVD including macrophages and how other cells respond to macrophage produced signaling.  We think including this information in one place is helpful to readers interested in this topic.

  • Several locations in the manuscript: “degenerate IVD” should be “degenerated IVD”.

We made this correction.

  • Figures 1 and 3 don’t not show what the figure legend describe. Please add additional detail to the figures. For Figure 3 it looks as if proteins/cytokines would be outside the IVD.

We revised these figures.

  • Table 1&2 should be removed. A table summarizing the literature about macrophages in disc degeneration would be more helpful.

We removed Table 2, but retained Table 1 for reasons given above.

Reviewer 3 Report

Comments to the Authors

Thank you for allowing me to review your paper. In this review, the authors discuss the hallmarks of IVD degeneration, showcase evidence of macrophage involvement during disc degeneration, and explore burgeoning research aimed at understanding the molecular pathways regulating macrophage functions during IVD degeneration. I am very impressed that the authors are researching macrophages, including IVD. However, it is a list of sentences, and it is not easy to understand what is essential. Please summarize the role of the macrophage in IVD degeneration in an easy-to-understand table or figure for the reader.

Below are my comments.

-Introduction is nothing new as it has already been reported in many papers. The description of macrophage is from lines 89 to 95 and should be discussed further in the introduction.

Please use tables and figures to help readers understand the role of macrophages in disc degeneration. It is a list of sentences, and it is difficult for me, including the reader, to understand it as a Review. The critical point in this paper is 4. Roles of Macrophages in Disc Degeneration, so please use sentences, tables, and figures about it.

Line 302-304, “Although a number of studies show positive correlation between 1) macrophages within the IVD, 2) disc degeneration, and 3) associated pain, we have much to learn about the control of macrophage function within the disc environment.” Which paper specifically?

Author Response

We thank the Reviewers for their critical and helpful review of our narrative review on Macrophages and Intervertebral Disc Degeneration.  Below we address each comment in a point-by-point fashion.

Reviewer 3

Thank you for allowing me to review your paper. In this review, the authors discuss the hallmarks of IVD degeneration, showcase evidence of macrophage involvement during disc degeneration, and explore burgeoning research aimed at understanding the molecular pathways regulating macrophage functions during IVD degeneration. I am very impressed that the authors are researching macrophages, including IVD. However, it is a list of sentences, and it is not easy to understand what is essential. Please summarize the role of the macrophage in IVD degeneration in an easy-to-understand table or figure for the reader.

Below are my comments.

-Introduction is nothing new as it has already been reported in many papers. The description of macrophage is from lines 89 to 95 and should be discussed further in the introduction.

An introduction sets the stage for the problem being discussed, casting a large picture of the problem (i.e., intervertebral disc degeneration), and then narrowing down to the specific topic being discussed (i.e., what is known about macrophages).  We added detail to our outline of how macrophages impact IVDD to our introduction as suggested.

Please use tables and figures to help readers understand the role of macrophages in disc degeneration. It is a list of sentences, and it is difficult for me, including the reader, to understand it as a Review. The critical point in this paper is 4. Roles of Macrophages in Disc Degeneration, so please use sentences, tables, and figures about it.

We modified our Figures as suggested.

Line 302-304, “Although a number of studies show positive correlation between 1) macrophages within the IVD, 2) disc degeneration, and 3) associated pain, we have much to learn about the control of macrophage function within the disc environment.” Which paper specifically?

This sentence summarizes most of the preceding references cited, so no one paper or handful of papers encapsulate this statement.  It is our thought.